# Solvent-triggered reversible interconversion of all-nitrogen-donor-protected silver nanoclusters and their responsive optical properties

Shang-Fu Yuan [1,2], Zong-Jie Guan[2], Wen-Di Liu[1] & Quan-Ming Wang [1,2]

Surface organic ligands are critical in determining the formation and properties of atomically precise metal nanoclusters. In contrast to the conventionally used thiolate, phosphine and alkynyl ligands, the amine ligand dipyridylamine is applied here as a protecting agent in the synthesis of atomically precise metal nanoclusters. We report two homoleptic amido-protected Ag nanoclusters as examples of all-nitrogen-donor-protected metal nanoclusters: $[Ag_{21}(dpa)_{12}]SbF_6$ (**Ag₂₁**) and $[Ag_{22}(dpa)_{12}](SbF_6)_2$ (**Ag₂₂**) (dpa = dipyridylamido). Single crystal X-ray structural analysis reveals that both clusters consist of a centered-icosahedron $Ag_{13}$ core wrapped by 12 dpa ligands. The flexible arrangement of the N donors in dpa facilitates the solvent-triggered reversible interconversion between **Ag₂₁** and **Ag₂₂** due to their very different solubility. The successful use of dpa in the synthesis of well-defined silver nanoclusters may motivate more studies on metal nanoclusters protected by amido type ligands.

[1] Department of Chemistry, Tsinghua University, 100084 Beijing, China. [2] Department of Chemistry, College of Chemistry and Chemical Engineering, Xiamen University, 361005 Xiamen, China. Correspondence and requests for materials should be addressed to Q.-M.W. (email: qmwang@tsinghua.edu.cn)

Along with the advance of synthetic methodologies of ligand passivated nanoparticles in the past decades[1], intense attention is also paid to synthesize atomically precise nanoclusters. The precise control over number of atoms and their arrangements within the clusters could provide valuable possibilities in modulating the various properties of nanoclusters[2–10]. The isolation of metal nanoclusters needs the protection from organic/inorganic, which play important roles in determining the stability, atom packing, and properties of metal clusters. Phosphine[11–15], thiolate[16–23], and alkynyl ligands[24–30] have been extensively used in the synthesis of metal nanoclusters, but the detailed amido-metal interfaces are still elusive due to the lack of total structural determination of amido-protected metal nanoclusters[31–33]. The unveiling of metal-amine interface will not only be helpful for understanding the formation of related clusters and establishing structure-property relationships, but also provides valuable information for the controlled synthesis of nanoparticles. In order to isolate a stable amido-protected metal nanocluster, which is helpful for making structural determination possible, our strategy is to use dipyridylamine (Hdpa) as a multidentate protecting agent. After deprotonation, Hdpa turns into a monoanionic dipyridylamido (dpa) ligand, which contains one amido and two pyridyl N donors available for bridging multi metal centers. The two pyridyl groups provide additional binding sites for metal atoms, which is advantageous in stabilizing metal clusters.

On the other hand, many factors have been found influencing the formation of metal clusters, such as ligand hindrance, temperature and pH values. Ligand hindrance has been demonstrated to control the size of $Au_n(SR)_m$ nanoclusters[34]. $Au_{130}(p\text{-}MBT)_{50}$, $Au_{104}(m\text{-}MBT)_{41}$, and $Au_{40}(o\text{-}MBT)_{24}$ nanoclusters were obtained with methylbenzenethiols (MBT) as a result of their different positions of the methyl groups. A metastable $Au_{38}(SC_2H_4Ph)_{24}$ isolated at low temperature could be irreversibly converted to biicosahedral isomer at 50 °C[19]. Captopril (Capt) ligand stabilized $Au_{25}Capt_{18}$ was transformed into highly fluorescent $Au_{23}Capt_{17}$ under treatment of HCl[35]. Solvents were also demonstrated to be important in some cases, e.g. an icosahedral-to-cuboctahedral structural transition was achieved by adding hexane into an ethanol solution of $Au_{13}(PPh_3)_4(SC_{12}H_{25})_4$ $(SC_{12}H_{25} = \text{dodecanethiol})$[36]. Herein, we report an interesting case that reversible interconversion of clusters can be triggered by different solvents.

With the protection of dpa, we are able to isolate all-amido-protected Ag nanoclusters: $[Ag_{21}(dpa)_{12}]SbF_6$ ($Ag_{21}$) and $[Ag_{22}(dpa)_{12}](SbF_6)_2$ ($Ag_{22}$). The total structural determination of the clusters have been carried out. To our surprise, it is found that $Ag_{21}$ and $Ag_{22}$ can be reversibly interconverted via solvent triggers, i.e., the transformation may be induced by solvents, due to their different solubility. Both clusters have very similar structures based on a centered-icosahedral $Ag_{13}$ core, with different number of shell silver atoms ligated by the same number of homoleptic dpa ligands. Although $Ag_{21}$ and $Ag_{22}$ have similar structures, they display significantly different optical properties.

## Results

### Synthesis
The preparation of $Ag_{21}$ involves the reduction of the dpa-Ag precursor in the presence of phosphine, Hdpa, $AgSbF_6$, and MeONa in dichloromethane. Similarly, the $Ag_{22}$ cluster can be obtained without addition of Hdpa, and the amount of $NaBH_4$ should be decreased (see Supporting Information). Although phosphines did not appear in the resulted clusters, they are important for the preparation. We monitored the $^{31}P$ spectra in the synthesis of $Ag_{22}$, a signal (9.76 ppm) was found at lower field in comparison with that of $[Ag_2(dppb)_2]^{2+}$ (−1.60 ppm) (dppb = 1,4-bis-(diphenylphosphino)butane). This fact indicates that

certain intermediates containing phosphines were formed in the reaction, which are helpful for the protection of silver ions (Supplementary Fig. 1). We also obtained crude samples of $Ag_{21}$ with either $Ph_3P$ or dppm (bis(diphenylphosphino)methane) instead of dppb (Supplementary Fig. 2). In addition, the reduction process could be largely inhibited under basic conditions[37]. It was known that the species reduces gold(I) is $BH_3OH^-$, which comes from the hydrolysis of $BH_4^-$[38]. The hydrolysis was catalyzed by $H^+$, so in basic solutions, the reduction process becomes more slowly[39]. A rapid reduction without MeONa gave a red solution that showed a broad plasmonic resonance absorption peak at ~450 nm, which indicates the formation of nanoparticles instead of clusters (Supplementary Fig. 3).

### Mass spectrometry and optical properties
The positive ESI-MS spectrum of $Ag_{21}$ shows the molecular ion peak at $m/z = 4307.71$, corresponding to the monocation $[Ag_{21}(dpa)_{12}]^+$ (calcd 4307.87) (Fig. 1a), and the spectrum of $Ag_{22}$ gave signal of a dication $[Ag_{22}(dpa)_{12}]^{2+}$ at $m/z = 2207.39$ (Fig. 1b). The observed isotopic patterns of the clusters are both in perfect agreement with their simulated. X-ray photoelectron spectroscopy (XPS) revealed that the $Ag3d_{5/2}$ binding energy in $Ag_{21}$ and $Ag_{22}$ were observed at 368.0 and 368.2 eV, respectively, which suggested that the oxidation state of Ag were closer to Ag(0) (367.9 eV) (Supplementary Fig. 4).

In $CH_2Cl_2$, $Ag_{21}$ is a brown red solution, while $Ag_{22}$ is green. The UV/Vis spectrum of $Ag_{21}$ shows optical absorption bands at 352, 422, 520 and 576 nm (shoulder), whereas $Ag_{22}$ has absorptions at 354, 437, 505 and 582 nm (Fig. 1c). In detail, the absorption peak in $Ag_{21}$ at 422 nm red shifts to 437 nm, while the peak at 520 nm blue shifts to 505 nm in $Ag_{22}$. Specifically, the first absorption peak at 576 nm in $Ag_{21}$ becomes more prominent in $Ag_{22}$ and red shifts to 582 nm. These spectral differences can also be observed in the photon-energy plots (Fig. 1d). By extrapolating the absorbance to zero, the optical energy gaps were determined to be 1.92 eV for $Ag_{21}$ and 1.80 eV for $Ag_{22}$, respectively. The difference in the optical properties of the two similar clusters indicates that the surface units have a significant disturbance to the electronic and optical properties[25,40].

### Molecular structures
Single crystal structural analysis revealed that $Ag_{21}$ comprises a monocation cluster $[Ag_{21}(dpa)_{12}]^+$ while the $Ag_{22}$ cluster contains a dication cluster $[Ag_{22}(dpa)_{12}]^{2+}$ of $C_3$ symmetry, both clusters have $SbF_6^-$ counteranions. Both clusters consist of a centered-icosahedron $Ag_{13}$ core, and the surrounding protection are provided by 12 dpa ligands. The $C_3$ axis passes through the central Ag atom and the centers of two (opposite) triangular faces of $Ag_{12}$ icosahedron (Fig. 2a, b). Eight Ag atoms caps the $Ag_3$ faces of the icosahedron in $Ag_{21}$. Two of them lie on the $C_3$ axis and the remaining six locate around the $C_3$ axis in three groups alongside the icosahedron (Fig. 2c, d). Thus, the ideal symmetry of the $Ag_{21}$ core is $D_{3h}$. Similarly, nine of the twenty $Ag_3$ faces of the icosahedron are capped by 9 Ag atoms to form a $Ag_{22}$ cluster and they are related by a $C_3$ axis (Fig. 2e). Among them, three are located at the bottom, and the remaining six Ag atoms form three $Ag_2$ pairs (Fig. 2f). In the $Ag_{13}$ core of $Ag_{21}$, the center-surface $Ag\cdots Ag$ distances are in the range of 2.757–2.832 Å with an average value of 2.790 Å, while the surface–surface $Ag\cdots Ag$ distances are in the range of 2.770–3.168 Å with an average value of 2.933 Å. The values are comparable with the $Ag\cdots Ag$ distances of icosahedron $Ag_{13}$ observed in $[Ag_{35}(H_2L)_2(L)(C\equiv CBu^t)_{16}](SbF_6)_3$ ($H_4L$, $p$-tert-butylthiacalix[4]-arene)[41] (2.778 and 2.923 Å). Moreover, the distances of capping Ag atoms to $Ag_{13}$ kernel have an average value of 2.867 Å, while the average distances between capping atoms are 3.054

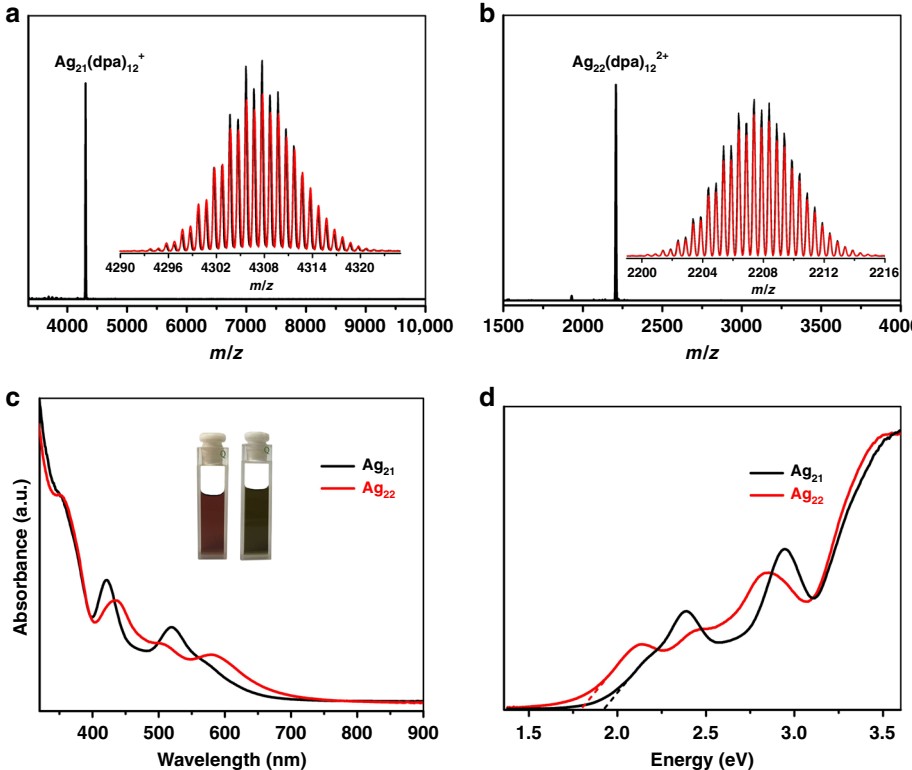

**Fig. 1** ESI mass and optical absorption spectra of **Ag21** and **Ag22** in CH2Cl2. Mass spectra of **Ag21** (**a**) and **Ag22** (**b**), inset: the measured (black trace) and simulated (red trace) isotopic distribution patterns of the corresponding the molecular ion peaks. **c** UV-vis absorption spectra with photographs of **Ag21** (left) and **Ag22** (right) under ambient light. **d** Photon-energy plot in CH2Cl2

Å, indicating a compact contact between capping atoms to the Ag13 kernel. Similar Ag···Ag bond distances were also observed in **Ag22** (Supplementary Table 1).

The main structural feature in the two Ag clusters is that various interfacial binding geometries generated by dpa ligands. The dpa ligand has three coordination donors (one amido N donor and two pyridine N donors). It function as efficient surface protecting ligands, and there are four types of binding motifs of dpa with silver atoms presented in **Ag21** and **Ag22** (Fig. 3). In **Ag21**, three dpa ligands can be viewed as simple bridging groups (motif A) and six dpa ligands adopt μ3-bridging mode coordinating three silver atoms with each N donor ligating a Ag (motif B) (Supplementary Fig. 5). Only motif B has been previously found in Ag cluster[42] and the two binding modes have been also reported in other transition metals (e.g., Cr, Mo)[43,44]. Interestingly, the amido N of dpa could also ligate two metal atoms (motif C and motif D), so that the maximum coordination number of dpa can be 4. The two binding modes have never been reported before. The icosahedral Ag12 cores of **Ag21** and **Ag22** are connected to three pyridine N atoms and nine amido N atoms (Supplementary Figs. 5 and 6). The 12 dpa ligands in **Ag22** adopt three different binding modes: six in motif B, three in motif C and three in motif D (Supplementary Fig. 6). The $N_{py}$···Ag bond lengths in **Ag21** are in the range of 2.108–2.371 Å and the $N_{amido}$···Ag bond lengths range from 2.042 to 2.510 Å. Similar values are also found in **Ag22**. (Supplementary Table 1).

The arrangements of the silver atoms and dpa ligands are shown in Fig. 4. The conversion from **Ag21** to **Ag22** can be structurally understood as following: with the addition of a silver atom (bright green) to the surface of **Ag21**, the two cap silver atoms (yellow) move far apart from each other and the other two (dark green) twist their orientation significantly (Fig. 4a vs 4b). The surface dpa ligands can be divided into four groups as

marked in different color in Fig. 4c, d. The positions of dpa ligands are just slightly changed, which allows the addition one silver ion to the Ag21 core or removal of one silver ion from the Ag22 core. Detailed surface structural comparison is presented in Supplementary Fig. 7.

**Interconversion.** The flexible arrangement of the three N donors of dpa makes it possible to adjust the coordination positions of dpa, which is an important prerequisite for cluster interconversion. It was found that there is a solvent-dependent equilibrium between **Ag21** and **Ag22** (Fig. 5a). The interconversion process can be monitored with absorption spectroscopy, because they have significantly different absorption profiles. The interconversion involves the adding or leaving of a Ag+ ion on the surface structure of the clusters. The single Ag-species should hardly affect the absorption spectra due to its d10 nature. As shown in Fig. 5b, **Ag22** dissolved in a mixture of EtOH and n-hexane (v: v = 1: 4) exhibits very similar profile to that of the freshly prepared **Ag21**. As the ratio of n-hexane to EtOH increases, the peak at 582 nm became weaker. Meanwhile, the peaks at 505 and 437 nm became stronger and they were shifted to 520 and 422 nm, respectively. These changes indicates that [Ag22(dpa)12]2+ has been gradually converted into [Ag21(dpa)12]+. It is worth mentioning that three isosbestic points were observed at ~565, 500 and 435 nm in the UV-vis absorption spectra, which indicates one-to-one conversion stoichiometry from **Ag22** to **Ag21**. Moreover, the **Ag22** species could be recovered by increasing the ratio of EtOH to n-hexane (Supplementary Fig. 8). Both clusters have identical icosahedral Ag13 cores, and the positions of their dpa ligands are just slightly changed. So, in the interconversion, the surface structure of the cluster only need to slightly adjust for the taking or releasing a Ag+ ion. It should be mentioned that both

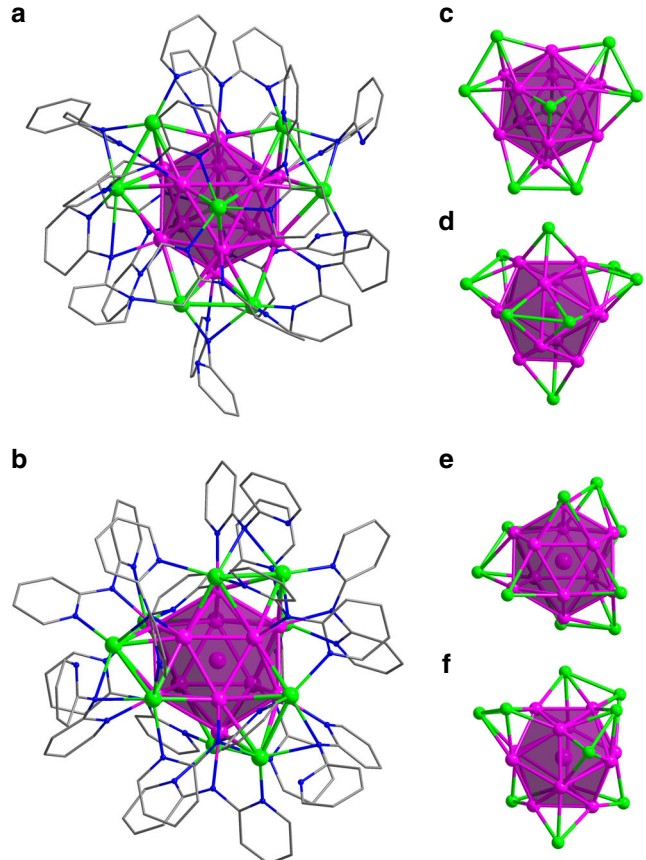

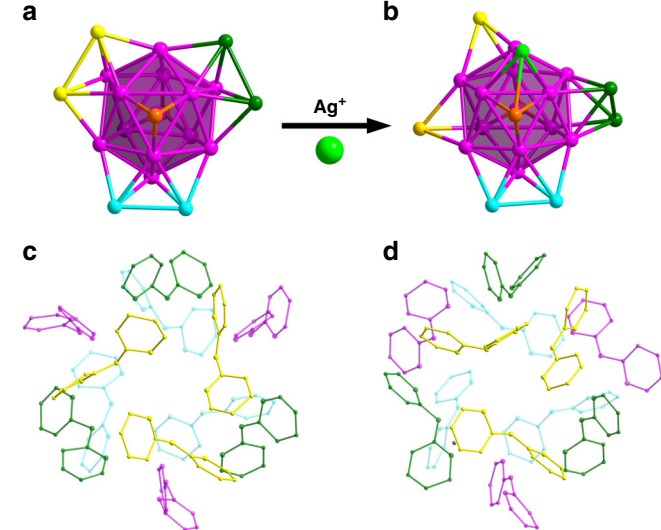

**Fig. 4** Structural comparison of **Ag₂₁** and **Ag₂₂**. Kernel structures (**a** vs **b**) and surface ligand arrangements (**c** vs **d**) of [Ag₂₁(dpa)₁₂]⁺ and [Ag₂₂(dpa)₁₂]²⁺

**Fig. 2** Molecular structures of **Ag₂₁** and **Ag₂₂**. Total structure of [Ag₂₁(dpa)₁₂]⁺ (**a**) and [Ag₂₂(dpa)₁₂]²⁺ (**b**) showing the Ag₁₃ polyhedron; Top view (**c**) and side view (**d**) of the Ag₂₁ kernel; Top view (**e**) and side view (**f**) of the Ag₂₂ kernel. Color codes: purple and green sphere, Ag; blue sphere, N; gray sphere, C

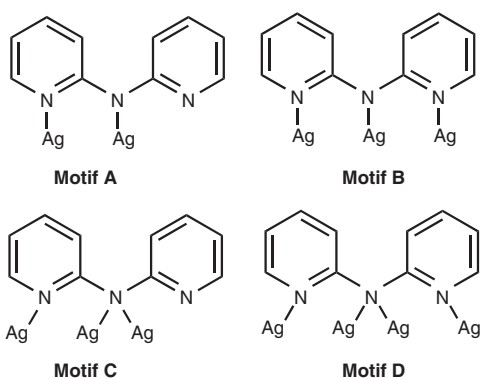

**Fig. 3** Schematic representation of the binding modes of dpa in **Ag₂₁** and **Ag₂₂**

**Ag₂₁** and **Ag₂₂** carry a valence electron count of 8e. The iso-electronic nature also contribute to the easy and reversible interconversion of these two nanoclusters.

The solvent-triggered interconversion was also confirmed by ESI-MS (Fig. 5d–f). When the ratio of EtOH to n-hexane was 1: 4, **Ag₂₁** species was detected as a prominent peak at $m/z = 4307.80$ ([Ag₂₁(dpa)₁₂]⁺, Fig. 5e), and the recovery of **Ag₂₂** from **Ag₂₁** was also verified by ESI-MS (Fig. 5f). The prominent peak at $m/z = 2207.26$ was assigned to [Ag₂₂(dpa)₁₂]²⁺, and the two small peaks

with lower m/z values are [Ag₂₁(dpa)₁₂]²⁺ at 2153.76 (calcd 2153.93) and [Ag₂₀(dpa)₁₂]²⁺ at 2099.27 (calcd 2099.48), which are fragment ions after losing one or two Ag(0). The small peak at 2253.27 is [Ag₂₂(dpa)₁₂(EtOH)₂]²⁺ (calcd 2253.43). Interestingly, the interconversion are also sensitive to temperature (Fig. 5c). Lower temperature such as −20 °C favors the formation of **Ag₂₂** species, and the reverse transformation to **Ag₂₁** happened when the temperature of the solution was elevated to room temperature.

Both **Ag₂₁** and **Ag₂₂** are soluble in ethanol, and the solubility of **Ag₂₁** is 14 times as much as that of **Ag₂₂**. As the arrangement of protecting ligands are very similar on the surface of both nanoclusters, the different solubility is largely due to that **Ag₂₁** is +1 charged and **Ag₂₂** carries two positive charges. Both clusters are not soluble in n-hexane, but **Ag₂₁** is still soluble in the mixed solution of ethanol/n-hexane due to its excellent solubility in ethanol. As for **Ag₂₂**, its solubility is significantly decreased as the increasing of the content of n-hexane in the mixed solution of ethanol/n-hexane, which forces the equilibrium moving to the formation of much more soluble **Ag₂₁**. Control experiments were carried out to make sure that there is no concentration effect. The UV-vis spectra of EtOH solutions containing different concentration of **Ag₂₁** or **Ag₂₂** were measured. As shown in Supplementary Fig. 9, the absorption peaks become weaker as the concentrations decrease, but no transformation occurred. Since the absorbance is proportional to the concentration of **Ag₂₁** or **Ag₂₂**, following the Lambert–Beer law, no aggregation occurs. Therefore, it is reasonable to conclude that the interconversion between **Ag₂₂** and **Ag₂₁** in solution is due to their largely different solubility. This statement is also supported by the temperature-dependent interconversion as above-mentioned (Fig. 5c). Recently, Xie et al. revealed a size conversion reaction from [Au₂₃(p-MBA)₁₆]⁻ to [Au₂₅(p-MBA)₁₈]⁻ (p-MBA = para-mercaptobenzoic acid), which could be induced by changing the solvent polarity (e.g., from water/ethanol to pure water)[45]. This work is certainly a different case.

## Discussion
In summary, we have synthesized two all-amido-protected metal nanoclusters, and achieved their total structure

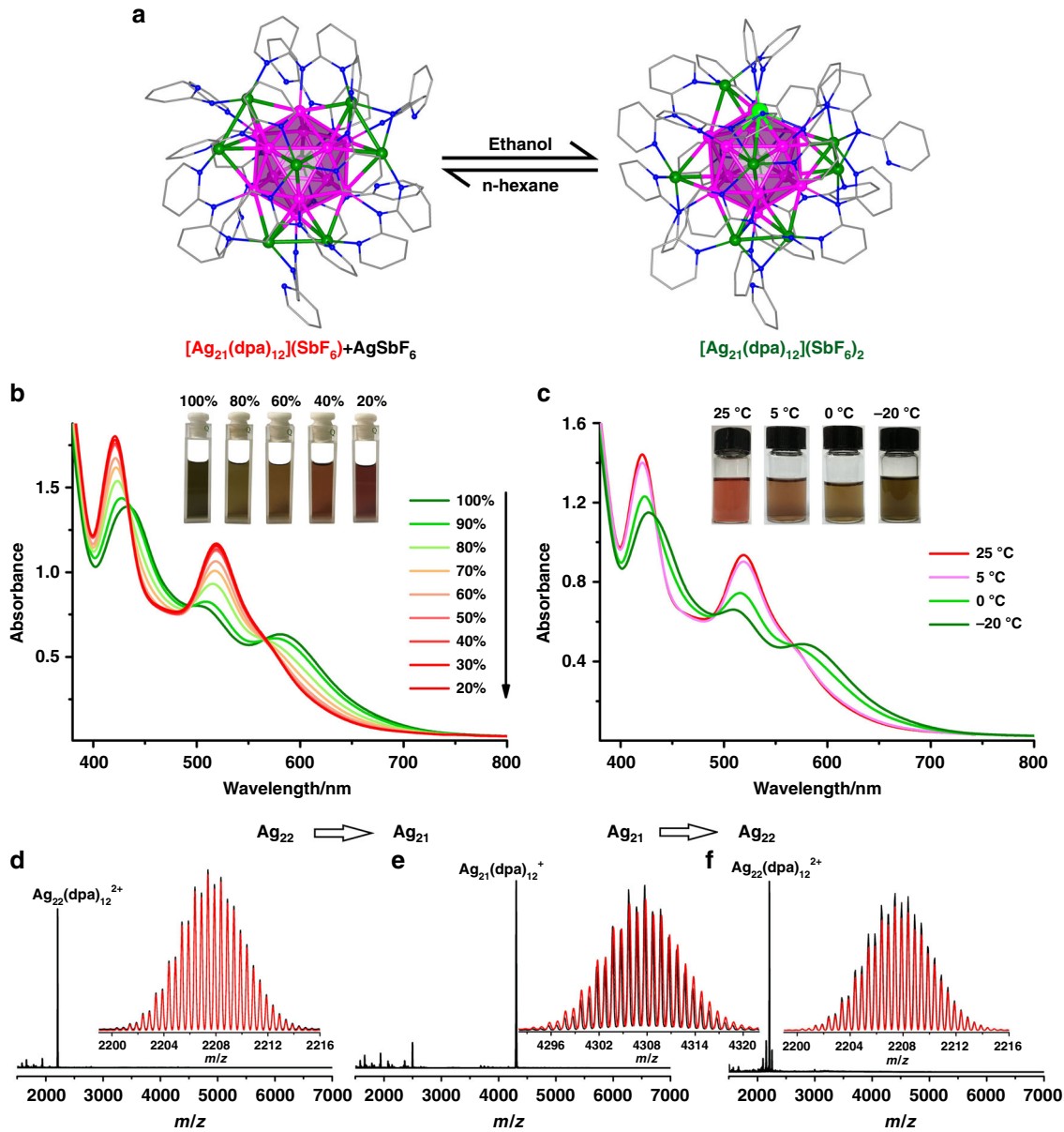

**Fig. 5** The reversible interconversion between **Ag₂₂** and **Ag₂₁**. **a** Solvent-dependent equilibrium between **Ag₂₁** and **Ag₂₂**. Color codes: purple, green, and bright green sphere, Ag; blue sphere, N; gray sphere, C. **b** UV-vis spectra of **Ag₂₂** dissolved in mixed solvents with various ratios of EtOH to n-hexane (the total volume of EtOH and n-hexane is kept the same). **c** UV-vis spectra of **Ag₂₂** dissolved in a mixture of EtOH and n-hexane (v : v = 1 : 4) at different temperatures. ESI-MS spectra: **d** **Ag₂₂** dissolved in EtOH. **e** Formation of **Ag₂₁** cluster from **Ag₂₂** when the volume ratio of EtOH to n-hexane is 1: 4. **f** Recovery of **Ag₂₂** from **Ag₂₁** when the volume ratio of EtOH and n-hexane is changed back to 9 : 1

determination. The flexible arrangement of the three N donors of dpa facilitates the formation of various interfacial Ag-N binding geometries, which favors the generation of cluster diversity. The solvent-triggered reversible interconversion presents an access for the atomic-level tailoring of the nanocluster structures and the regulation of their properties. The role of solvents on the transformation of cluster species and even nanoparticles is worth noting, which presents possibilities in obtaining desired clusters or particles in a convenient way. On the other hand, four binding modes of the dpa ligand have been revealed, and the coordination of dpa highlights its promising potential to act as a protecting agent in the formation of metal nanoclusters. This work sheds light on understanding of the metal−amido interface and the mechanism of the cluster conversion, which will stimulate

the investigation on metal nanoclusters or even nanoparticles containing ligands of various amines.

## Methods

**Chemicals and materials**. In all, 2, 2′-dipyridylamine, Sodium methanolate (MeONa, 98%) and Silver hexafluoroantimonate (AgSbF₆, 98.0%) were purchased from J&K; sodium borohydride (NaBH₄, 98%) and other reagents employed were purchased from Sinopharm Chemical Reagent Co. Ltd. (Shanghai, China). Other reagents employed were commercially available and used as received.

**Synthesis of dpaAg**. To a solution of AgNO₃ (500 mg, 2.94 mmol) in H₂O (4 mL) and MeOH (100 mL), a solution of Hdpa (554 mg, 3.2 mmol) in MeOH (10 mL) was added under stirring. Then 612 μL of NEt₃ (446 mg, 4.46 mmol) was added to the solution, a light yellow precipitate was produced. The mixture was stirred for 30 min at room temperature in air in the dark. Then the solution was filtered to give a light yellow solid, which was washed with water (2 × 5 mL), ethanol (5 mL) and ether (5 mL) to give 728.6 mg (89% yield based on Ag) of dpaAg.

**Synthesis of Ag$_{21}$(dpa)$_{12}$(SbF$_6$) (Ag$_{21}$).** To 2 mL CH$_2$Cl$_2$ suspension containing dpaAg (27 mg, 0.1 mmol), dppb (8 mg, 0.019 mmol), and Hdpa (17 mg, 0.1 mmol), and 1 mL AgSbF$_6$ (5 mg) in MeOH and 1 mL 2 M MeONa (11 mg, 0.2 mmol) in MeOH were added. After stirring for 30 min, a freshly prepared solution of NaBH$_4$ (0.15 mg in 1.0 mL of ethanol) was added dropwise with vigorous stirring. The solution color changed from light yellow to reddish-brown, and the reaction continued overnight. Then the reaction was evaporated to dryness to give a reddish-brown solid. The solid was washed with n-hexane (2 × 3 mL), then dissolved in 3 mL CH$_2$Cl$_2$: toluene (v : v = 3 : 1), and the resulted solution was centrifuged for 3 min at 10,000 r.min$^{-1}$. The reddish-brown supernatant was collected and subjected to vapor diffusion with ether: n-hexane (v : v = 1 : 5) to afford reddish-brown crystals after 7 days in 5.2 mg yield (24% based on Ag).

Anal. UV-vis (λ, nm): 352; 422; 520; 576 (shoulder peak). (E, eV): 3.54; 2.94; 2.39; 2.17. Eg = 1.92 eV. ESI-MS (CH$_2$Cl$_2$): 4307.71 ([Ag$_{21}$(dpa)$_{12}$]$^+$). XPS (binding energy, eV): Ag 3d$_{5/2}$, 368.0; Ag 3d$_{3/2}$, 374.0 eV.

**Synthesis of Ag$_{22}$(dpa)$_{12}$(SbF$_6$)$_2$ (Ag$_{22}$).** To 2 mL CH$_2$Cl$_2$ suspension containing dpaAg (27 mg, 0.1 mmol) and dppb (8 mg, 0.019 mmol), 1 mL AgSbF$_6$ (5 mg) in MeOH and 1 mL 2 M MeONa (11 mg, 0.2 mmol) in MeOH were added. After stirring for 30 min, a freshly prepared solution of NaBH$_4$ (0.10 mg in 1.0 mL of ethanol) was added dropwise with vigorous stirring. The solution color changed from light yellow to green, and the reaction continued overnight. Then the reaction was evaporated to dryness to give a green solid. The solid was washed with ether: n-hexane (v : v = 1 : 3) (5 mL), then dissolved in 3 mL CH$_2$Cl$_2$: toluene (v : v = 3 : 1), and the resulted solution was centrifuged for 3 min at 10,000 r.min$^{-1}$. The green supernatant was collected and subjected to vapor diffusion with ether: n-hexane (v : v = 1 : 5) to afford black crystals after 10 days in 5 mg yield (20% based on Ag).

Anal. UV-vis (λ, nm): 354; 437; 505; 582. (E, eV): 3.52; 2.85; 2.44; 2.13. Eg = 1.80 eV. ESI-MS (CH$_2$Cl$_2$): 2207.41 ([Ag$_{22}$(dpa)$_{12}$]$^{2+}$). XPS (binding energy, eV): Ag 3d$_{5/2}$, 368.2; Ag 3d$_{3/2}$, 374.2 eV.

**General procedure for transformation.** 20 mg Ag$_{22}$ was dissolved in 1 mL EtOH, and the solution was divided into 10 portions of 30 μL. Then, different volume of EtOH and n-hexane were added, and make sure the total volume is 3 mL. With the increase of the ratio of n-hexane, the color of the solution gradually changed from green to red. For the reverse transformation, different volume of EtOH and n-hexane were added to the obtained red solution in the same method. With the increase of the ratio of EtOH, the red solution gradually return back to green. The transformation was very fast, which was monitored by measuring UV-Vis absorption spectra of the mixture after gentle shaking.

**Physical measurements.** UV-Vis absorption spectra was recorded on a Shimadzu UV-2550 Spectrophotometer. Mass spectrum was recorded on an Agilent Technologies ESI-TOF-MS. Fourier-transform infrared spectroscopy (FT-IR) spectra were collected in the range of 4000–400 cm$^{-1}$ with a Bruker FT-IR spectrometer. Nuclear magnetic resonance (NMR) data were collected on a JEOL ECS-400 spectrometer. X-ray photoelectron spectroscopy (XPS) studies were performed on PHI Quantum-2000 XPS. The sample was put under UHV to reach the 10$^{-8}$ Pa range. The non-monochromatized Al Kα source was used at 10 kV and 10 mA. All binding energies were calibrated using the C (1s) carbon peak (284.6 eV), which was applied as an internal standard. High resolution narrow-scan spectra were recorded with the electron pass energy of 50 eV and takeoff angle of 55° to achieve the maximum spectral resolution.

**X-ray crystallography.** Intensity data of **Ag$_{21}$** and **Ag$_{22}$** were collected on an Agilent SuperNova Dual system (Cu Kα) at 180 K and 173 K, respectively. Absorption corrections were applied by using the program CrysAlis (multi-scan) and all the structures were solved by direct methods. All non-hydrogen atoms of **Ag$_{22}$** were refined anisotropically by least-squares on F$_2$ using the SHELXTL program. The hydrogen atoms of organic ligands were generated geometrically. Ag atoms of **Ag$_{21}$** were refined with anisotropic displacement parameters, while others were refined with isotropic atomic displacement parameters. SQUEEZE routine in PLATON was employed in the structural refinements.

## Data availability

The data that support the findings of this study are available from the corresponding author upon reasonable request. The X-ray crystallographic coordinates for structures reported in this article (see Supplementary Tables 2 and 3) have been deposited at the Cambridge Crystallographic Data Centre (CCDC) under deposition numbers CCDC 1902596 (**Ag$_{21}$**) and CCDC 1902597 (**Ag$_{22}$**). These data can be obtained free of charge from the Cambridge Crystallographic Data Centre via http://www.ccdc.cam.ac.uk/data_request/cif.

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

## Acknowledgements

This work was supported by the National Natural Science Foundation of China (21631007).

## Author contributions

Q.-M.W. proposed the research direction and guided the whole experiment. S.-F.Y. and W.-D.L. conducted the synthesis and characterization. Z.-J.G. assisted analyzing the data. S.-F.Y. and Q.-M.W. wrote the manuscript.

## Additional information

**Competing interests:** The authors declare no competing interests.

