## [Peer Review File · Nature Communications]

Reviewers' Comments:

Reviewer #1:

Remarks to the Author:

This manuscript reports synthesis, total structure determination and interconversion chemistry of a pair of all-nitrogen-donor-protected silver nanoclusters (NCs), i.e., $[\text{Ag}_{21}(\text{dpa})_{12}]^+$ and $[\text{Ag}_{22}(\text{dpa})_{12}]^{2+}$ (dpa = dipyridylamido ligand). As-produced $[\text{Ag}_{21}(\text{dpa})_{12}]^+$ and $[\text{Ag}_{22}(\text{dpa})_{12}]^{2+}$ NCs were carefully synthesized, and comprehensively characterized by UV-vis absorption spectroscopy, electrospray ionization mass spectrometry (ESI-MS), X-ray photoelectron spectroscopy (XPS) and X-ray crystallography. The most interesting findings are diverse binding modes of dpa ligands (Motifs A-D in Figure 3) and the solvent-induced interconversion of aforementioned two NCs. The diversity of binding modes originate from the multi-dentate nature (μ_2 - μ_4) of dpa ligands, while such diversity in binding modes gives rise to reversible conversion between $[\text{Ag}_{21}(\text{dpa})_{12}]^+$ and $[\text{Ag}_{22}(\text{dpa})_{12}]^{2+}$ at changed solvent polarities. Exploring the binding chemistry between noble metal atoms and organic ligands, as well as interconversion pathways of atomically precise metal NCs, are of core importance in the development of total synthesis chemistry for functional inorganic nanomaterials at atomic precision. Therefore, I believe this well-executed work could appeal to heterogeneous readers of Nature Communications in the fields of nanoscience, supramolecular chemistry and materials science & engineering. I would like to recommend acceptance of this well-written manuscript after minor revisions (see my detailed comments below).

1. I noted that phosphine ligands are indispensable in the synthesis of either $[\text{Ag}_{21}(\text{dpa})_{12}]^+$ or $[\text{Ag}_{22}(\text{dpa})_{12}]^{2+}$. However, such ligands are somehow absent in the product NCs. The authors attributed such absence to intermediating effects of phosphine ligands (Page 2, Lines 5-6). Do the authors have detailed formula/structure information about the phosphine-containing intermediates?
2. The complete interconversion of $[\text{Ag}_{21}(\text{dpa})_{12}]^+$ and $[\text{Ag}_{22}(\text{dpa})_{12}]^{2+}$ triggered by solvent polarity is fascinating. The authors attributed such interconversion to the varied solubility of $[\text{Ag}_{21}(\text{dpa})_{12}]^+$ and $[\text{Ag}_{22}(\text{dpa})_{12}]^{2+}$ in a dual-solvent system of ethanol and hexane. This is somehow out of my expectation, as the arrangement of protecting ligands are very similar on the surface of both NCs (Figure 4c and 4d). Could the authors comment on the origin of the altered solubility?
3. Both $[\text{Ag}_{21}(\text{dpa})_{12}]^+$ or $[\text{Ag}_{22}(\text{dpa})_{12}]^{2+}$ carry a valence electron count of 8 e⁻. The isoelectronic nature should also contribute to the ease and reversibility of interconversion reaction of these two NCs. The authors may like to discuss this aspect in the revised manuscript.
4. How is the long-term stability of $[\text{Ag}_{21}(\text{dpa})_{12}]^+$ and $[\text{Ag}_{22}(\text{dpa})_{12}]^{2+}$ under optimised solvent conditions? The stability may dictate their acceptance in practical applications.
5. There are several small peaks around the cluster peak of $[\text{Ag}_{22}(\text{dpa})_{12}]^{2+}$ in the ESI-MS spectrum of $[\text{Ag}_{22}(\text{dpa})_{12}]^{2+}$ recovered from $[\text{Ag}_{21}(\text{dpa})_{12}]^+$ (Figure 5e). What are they?
6. Color code should be specified for Supplementary Figure 4-6.

Reviewer #2:

Remarks to the Author:

This is a great piece of work by Wang and coworkers which describes the synthesis, structural and complete spectroscopic and physicochemical characterization of the first family of homoleptic amido-protected Ag nanoclusters. I think this is an important breakthrough in silver cluster field because N-

ligand was not thought as a preferred ligand to stabilize silver nanoclusters previously. Although they have the similar Ag₁₃ core, the optical properties are dramatically different. More interesting is the reversible transformation between Ag₂₁ and Ag₂₂ induced by solvent. I definitely find this work suitable for NatureComm and my recommendation is acceptance after the authors address the below comments:

- (1) The necessity of using DpaAg. Have authors tried to directly use AgNO₃ and Hdpa in the syntheses of two clusters?
- (2) In Figure 4, from a to b, the one more Ag atom should be highlighted such as using "Ag⁺ (green ball)" instead of "Ag⁺".
- (3) AgSbF₆ is the most expensive among all silver salts, why authors selected it instead of other common silver salts such as AgOAc, CF₃COOAg, CF₃SO₃Ag?
- (4) Some new advances in silver clusters should be noted such as Acc. Chem. Res. 2018, 51, 3104, Angew. Chem. Int. Ed. 2019, 58, 195 and J. Am. Chem. Soc. 2019, 141, 4460.
- (5) The IR should be provided for two new silver clusters.

Reviewer #3:

Remarks to the Author:

Ligand protected nanoparticles is being a focus of research for the last 2-3 decades but in the recent times the attention is shifted towards synthesizing nanoclusters with precise control over number of atoms and their arrangements within the clusters. Authors of the manuscript have published a fine article in Angew Chem recently regarding synthesis of Au-32 clusters. In the manuscript the authors have not only shown the synthesis of Ag-21 and Ag-22 clusters but also shown the reversible interconversion between them which is successfully triggered by solvents. This kind of control over the interconversion and, hence, the control over optical properties can be very useful for several chemical and biological applications. The work presented in this manuscript is very much original and is of interest to a wide audience in the field chemistry and material science. The presentation of the data is expertly done and the manuscript is written very concisely with necessary references. Although the work is praise worthy in many aspects, in order to publish it in a journal of Nature Publications Group I think some more work needs to be done as follows:

- 1) The nature journals are mainly for very broad audiences. Hence the work presented needs to be discussed in a broader perspective for general audience. I completely agree that the material presented in the manuscript is for wider audience but it was not written in that respect. My suggestions to the authors would be to elaborate on the importance of the work and its findings in terms of the current state of the affairs regarding the particle synthesis and its control. Also, how the results can influence the current understanding of the chemistry of cluster synthesis should be discussed in general terms.
- 2) Although authors have discussed the method of the synthesis to obtain highly monodispersed clusters of Ag-21 and Ag-22, some discussions should be presented regarding the chemistry of the process which inhibits the formation of the nanoparticles in the presence of MeONa.
- 3) As per the interconversion between Ag-21 and Ag-22, one Ag-atom needs to leave or enter the cluster, respectively. How does the Ag-atom exchange happen? How the presence of entities in the solution with single Ag-species effect the absorbance spectra? Authors should put some discussion about the interconversion process.

Hence, with these points I would like to look at the revised version of the manuscript before I accept it for publication in Nature Communications.

Response to reviewers

Reviewer #1:

This manuscript reports synthesis, total structure determination and interconversion chemistry of a pair of all-nitrogen-donor-protected silver nanoclusters (NCs), i.e., $[\text{Ag}_{21}(\text{dpa})_{12}]^+$ and $[\text{Ag}_{22}(\text{dpa})_{12}]^{2+}$ (dpa = dipyridylamido ligand). As-produced $[\text{Ag}_{21}(\text{dpa})_{12}]^+$ and $[\text{Ag}_{22}(\text{dpa})_{12}]^{2+}$ NCs were carefully synthesized, and comprehensively characterized by UV-vis absorption spectroscopy, electrospray ionization mass spectrometry (ESI-MS), X-ray photoelectron spectroscopy (XPS) and X-ray crystallography. The most interesting findings are diverse binding modes of dpa ligands (Motifs A-D in Figure 3) and the solvent-induced interconversion of aforementioned two NCs. The diversity of binding modes originate from the multi-dentate nature (μ_2 - μ_4) of dpa ligands, while such diversity in binding modes gives rise to reversible conversion between $[\text{Ag}_{21}(\text{dpa})_{12}]^+$ and $[\text{Ag}_{22}(\text{dpa})_{12}]^{2+}$ at changed solvent polarities. Exploring the binding chemistry between noble metal atoms and organic ligands, as well as interconversion pathways of atomically precise metal NCs, are of core importance in the development of total synthesis chemistry for functional inorganic nanomaterials at atomic precision. Therefore, I believe this well-executed work could appeal to heterogeneous readers of Nature Communications in the fields of nanoscience, supramolecular chemistry and materials science & engineering. I would like to recommend acceptance of this well-written manuscript after minor revisions (see my detailed comments below).

Comment: I noted that phosphine ligands are indispensable in the synthesis of either $[\text{Ag}_{21}(\text{dpa})_{12}]^+$ or $[\text{Ag}_{22}(\text{dpa})_{12}]^{2+}$. However, such ligands are somehow absent in the product NCs. The authors attributed such absence to intermediating effects of phosphine ligands (Page 2, Lines 5-6). Do the authors have detailed formula/structure information about the phosphine-containing intermediates?

Response: Thank you very much for the comments and suggestions. Our attempts to

prepare the title clusters without adding phosphine ligands were unsuccessful, and a large amount of intractable precipitates formed during the reduction process. When the same reaction was run in the presence of phosphines, the initial suspension turned into a pale yellow solution after 30 min stirring, which indicated the formation of certain intermediates, being helpful for the protection of silver ions. Taking the synthesis of **Ag₂₂** as an example, we monitored the ³¹P spectra of the reaction solution. As shown in Supplementary Figure 1, the free dppb in CH₂Cl₂ showed one peak at -11.32 ppm, and the obtained solution gave an obvious peak at 9.76 ppm before addition of NaBH₄. The resulted solution showed a prominent peak at -1.60 ppm after reduction with NaBH₄ overnight, which is corresponding to the [Ag₂(dppb)₂]²⁺. Considering the signal before reduction is at low field, one might suggest that the intermediate is a multinuclear compound containing phosphines, we are not able to have a detailed structure for it though. Upon reduction, the intermediate was converted to dppb₂Ag₂²⁺ complex as a result the formation of **Ag₂₂**.

The following figure has been added in supporting information as Supplementary Figure. 1.

Supplementary Figure. 1. ³¹P NMR spectra regarding the synthesis of **Ag₂₂**. (a) Free dppb in CH₂Cl₂. (b) before adding NaBH₄. (c) after overnight reduction with NaBH₄.

Comment: The complete interconversion of $[\text{Ag}_{21}(\text{dpa})_{12}]^+$ and $[\text{Ag}_{22}(\text{dpa})_{12}]^{2+}$ triggered by solvent polarity is fascinating. The authors attributed such interconversion to the varied solubility of $[\text{Ag}_{21}(\text{dpa})_{12}]^+$ and $[\text{Ag}_{22}(\text{dpa})_{12}]^{2+}$ in a dual-solvent system of ethanol and hexane. This is somehow out of my expectation, as the arrangement of protecting ligands are very similar on the surface of both NCs (Figure 4c and 4d). Could the authors comment on the origin of the altered solubility?

Response: $[\text{Ag}_{21}(\text{dpa})_{12}]^+$ is monocation, while $[\text{Ag}_{22}(\text{dpa})_{12}]^{2+}$ is a dication. The charge of the two compounds is different, which could be the origin of the difference in solubility.

Comment: Both $[\text{Ag}_{21}(\text{dpa})_{12}]^+$ or $[\text{Ag}_{22}(\text{dpa})_{12}]^{2+}$ carry a valence electron count of 8 e⁻. The isoelectronic nature should also contribute to the ease and reversibility of interconversion reaction of these two NCs. The authors may like to discuss this aspect in the revised manuscript.

Response: Thanks for reviewer's kind advice. We added "It should be mentioned that both **Ag₂₁** and **Ag₂₂** carry a valence electron count of 8 e. The isoelectronic nature also contribute to the easy and reversible interconversion of these two nanoclusters" in the discussion of interconversion.

Comment: How is the long-term stability of $[\text{Ag}_{21}(\text{dpa})_{12}]^+$ and $[\text{Ag}_{22}(\text{dpa})_{12}]^{2+}$ under optimised solvent conditions? The stability may dictate their acceptance in practical applications.

Response: Under ambient conditions, **Ag₂₁** and **Ag₂₂** are stable in CH_2Cl_2 for about 1 day, and they can be stored for more than one week at -20 °C.

Comment: There are several small peaks around the cluster peak of $[\text{Ag}_{22}(\text{dpa})_{12}]^{2+}$ in the ESI-MS spectrum of $[\text{Ag}_{22}(\text{dpa})_{12}]^{2+}$ recovered from $[\text{Ag}_{21}(\text{dpa})_{12}]^+$ (Figure 5e). What are they?

Response: The two small peaks with lower m/Z values are $[\text{Ag}_{21}(\text{dpa})_{12}]^{2+}$ at 2153.76 (calcd 2153.93) and $[\text{Ag}_{20}(\text{dpa})_{12}]^{2+}$ at 2099.27 (calcd 2099.48), which are fragment

ions after losing one or two Ag(0). The small peak at 2253.27 is $[\text{Ag}_{22}(\text{dpa})_{12}(\text{EtOH})_2]^{2+}$ (calcd 2253.43).

Comment: Color code should be specified for Supplementary Figure 4-6.

Response: Thanks for referee's kind reminder. Color codes have been added in the corresponding legends.

Reviewer #2:

This is a great piece of work by Wang and coworkers which describes the synthesis, structural and complete spectroscopic and physicochemical characterization of the first family of homoleptic amido-protected Ag nanoclusters. I think this is an important breakthrough in silver cluster field because N-ligand was not thought as a preferred ligand to stabilize silver nanoclusters previously. Although they have the similar Ag₁₃ core, the optical properties are dramatically different. More interesting is the reversible transformation between Ag₂₁ and Ag₂₂ induced by solvent. I definitely find this work suitable for Nature Communications and my recommendation is acceptance after the authors address the below comments.

Comment: The necessity of using dpaAg. Have authors tried to directly use AgNO₃ and Hdpa in the syntheses of two clusters?

Response: Thank you very much for the comments and suggestions, we had tried the direct reduction of Ag⁺ precursors in the presence of Hdpa, but large quantity of precipitates formed upon reduction.

Comment: In Figure 4, from a to b, the one more Ag atom should be highlighted such as using “Ag⁺ (green ball)” instead of “Ag⁺”.

Response: We have revised Figure 4 accordingly.

Comment: AgSbF₆ is the most expensive among all silver salts, why authors selected it instead of other common silver salts such as AgOAc, CF₃COOAg, CF₃SO₃Ag?

Response: As Sb is a heavy atom, SbF₆⁻ counteranions can be easily identified with single crystal structure analysis.

Comment: Some new advances in silver clusters should be noted such as *Acc. Chem. Res.* 2018, 51, 3104, *Angew. Chem. Int. Ed.* 2019, 58, 195 and *J. Am. Chem. Soc.* 2019, 141, 4460.

Response: These references were added as ref 9, ref 22 and ref 28.

Comment: The IR should be provided for two new silver clusters.

Response: IR spectra have been added as Supplementary Figure 10.

Reviewer #3:

Ligand protected nanoparticles is being a focus of research for the last 2-3 decades but in the recent times the attention is shifted towards synthesizing nanoclusters with precise control over number of atoms and their arrangements within the clusters. Authors of the manuscript have published a fine article in *Angew. Chem.* recently regarding synthesis of **Au₃₂** clusters. In the manuscript the authors have not only shown the synthesis of **Ag₂₁** and **Ag₂₂** clusters but also shown the reversible interconversion between them which is successfully triggered by solvents. This kind of control over the interconversion and, hence, the control over optical properties can be very useful for several chemical and biological applications. The work presented in this manuscript is very much original and is of interest to a wide audience in the field chemistry and material science. The presentation of the data is expertly done and the manuscript is written very concisely with necessary references.

Although the work is praise worthy in many aspects, in order to publish it in a journal of Nature Publications Group, I think some more work needs to be done as follows.

Comment: The nature journals are mainly for very broad audiences. Hence the work presented needs to be discussed in a broader perspective for general audience. I completely agree that the material presented in the manuscript is for wider audience but it was not written in that respect. My suggestions to the authors would be to elaborate on the importance of the work and its findings in terms of the current state of the affairs regarding the particle synthesis and its control. Also, how the results can influence the current understanding of the chemistry of cluster synthesis should be discussed in general terms.

Response: Thanks for the constructive suggestion, and we revised the beginning of introduction as: Along with the advance of synthetic methodologies of ligand passivated nanoparticles in the past decades, intense attention is also paid to synthesize atomically precise nanoclusters. The precise control over number of atoms and their arrangements within the clusters could provide valuable possibilities in

modulating the various properties of nanoclusters...The unveiling of gold-amine interface will not only be helpful for understanding the formation of related clusters and establishing structure-property relationships, but also provides valuable information for the controlled synthesis of nanoparticles.

Comment: Although authors have discussed the method of the synthesis to obtain highly monodispersed clusters of **Ag₂₁** and **Ag₂₂**, some discussions should be presented regarding the chemistry of the process which inhibits the formation of the nanoparticles in the presence of MeONa.

Response: It was known that the species reduces gold(I) is BH_3OH , which comes from the hydrolysis of BH_4^- . The hydrolysis was catalyzed by H^+ , so in basic solutions, the reduction process becomes more slowly (ref. Y. Okinaka, *J. Electrochem. Soc.: Electrochemical Science and Technology*, 1973, 739).

Comment: As per the interconversion between **Ag₂₁** and **Ag₂₂**, one Ag-atom needs to leave or enter the cluster, respectively. How does the Ag-atom exchange happen? How the presence of entities in the solution with single Ag-species effect the absorbance spectra? Authors should put some discussion about the interconversion process.

Response: Both clusters have identical icosahedral Ag_{13} cores, and the positions of their dpa ligands are just slightly changed. So, in the interconversion, the surface structure of the cluster only need to slightly adjust for the taking or releasing a Ag^+ ion. It should be mentioned that both **Ag₂₁** and **Ag₂₂** carry a valence electron count of 8 e. The isoelectronic nature also contribute to the easy and reversible interconversion of these two nanoclusters. The single Ag-species should hardly affect the absorption spectra due to its d^{10} nature. As shown in Fig. 5a, **Ag₂₂** dissolved in a mixture of EtOH and n-hexane ($v : v = 1 : 4$) exhibits very similar profile to that of the freshly prepared **Ag₂₁**.

Reviewers' Comments:

Reviewer #1:

Remarks to the Author:

The authors have addressed all my concerns, and I would like to suggest the acceptance of this nice paper.

Reviewer #3:

Remarks to the Author:

Glad to see all the revisions made by the authors. I am satisfied with the revised version and recommend for publication.